# Changes in Vitamin D Status in Korean Adults during the COVID-19 Pandemic

**DOI:** 10.3390/nu14224863

**Published:** 2022-11-17

**Authors:** Ji-Young Kwon, Sung-Goo Kang

**Affiliations:** Department of Family Medicine, St. Vincent’s Hospital, College of Medicine, The Catholic University of Korea, Seoul 16247, Republic of Korea

**Keywords:** COVID-19, social isolation, pandemic, SARS-CoV-2, coronavirus, vitamin D

## Abstract

The aim of this study was to investigate changes in 25(OH)D (25-hydroxyvitamin D) levels and in the vitamin D status of Korean adults before and during the coronavirus disease (COVID-19) pandemic. This study compared serum 25(OH)D levels before and after the pandemic in 1483 adults aged 19 years and older who were screened at a university hospital. Subjects were selected only from participants tested in the same season before and after the pandemic. The pre-COVID-19 testing period was from 1 March 2018 to 31 November 2019; the testing period in the COVID-19 era was from 1 June 2020 to 31 November 2021. The mean 25(OH)D level for all participants was 21.4 ± 10.2 ng/mL prior to the outbreak of COVID-19, which increased to 23.6 ± 11.8 ng/mL during the COVID-19 lockdown period (*p* < 0.001). The increase was particularly dramatic in elderly females (28.8 ± 12.3 ng/mL to 37.7 ± 18.6 ng/mL, *p* = 0.008). The prevalence of vitamin D deficiency decreased in both males (48.4% to 44.5%, *p* = 0.005) and females (57.0% to 46.0%, *p* < 0.001). In conclusion, 25(OH)D levels in Korean adults increased during the COVID-19 era, and the prevalence of vitamin D deficiency decreased accordingly.

## 1. Introduction

Since 11 March 2020, when the coronavirus disease (COVID-19) pandemic was first declared, the infection has continued to spread [1]. Fortunately, the rate of the spread of COVID-19 infection in South Korea (hereafter, Korea) is slowing [2]. The lockdown policy, which has been stringent until now, is being relaxed in line with this trend. In Korea, the outdoor mask-wearing mandate was rescinded on 2 May 2022 [3].

After the WHO’s declaration of the COVID-19 pandemic, nationally enforced lockdown orders were phased in according to the risk of spread. For more than two years, Koreans faced restrictions on leaving their homes and participating in social activities. Community health centers were closed, and telecommuting hours increased. Koreans prepared meals at home whenever possible, and face-to-face meetings and gatherings were reduced [4,5,6]. We assumed that these policies would have caused a decrease in vitamin D in the population due to reduced exposure to sunlight. Some related studies have suggested a decrease in population vitamin D levels [7,8], while others have reported no significant correlation between the COVID-19 pandemic and vitamin D levels [4,9,10]. Thus, the subject remains controversial.

Vitamin D helps maintain the calcium balance in the body. It also plays a significant role in bone metabolism [11]. Although vitamin D is obtained through the ingestion of some foods, the majority is synthesized when skin is exposed to the ultraviolet B radiation in sunlight [12,13,14]. It plays a meaningful role in preventing infection by bacteria and viruses [15,16,17]. It is also associated with several chronic and immune disorders, including obesity, diabetes, asthma, and atopic dermatitis [18,19,20,21,22,23]. Moreover, several studies have demonstrated its protective role against infection and exacerbation of COVID-19 [24,25]. Research findings showing that supplementation with vitamin D prevents respiratory infections [26] has sparked further interest in people’s vitamin D status after the COVID-19 pandemic.

There have been studies on the vitamin D deficiency status of people in the United States, Canada, and Europe. When the cutoff level of serum 25(OH)D (25-hydroxyvitamin D) was set below 50 nmol/L (20 ng/mL), the prevalence of vitamin D deficiency in these regions was 24.0%, 36.8%, and 40.4%, respectively [27]. In Korea, a study on the change in vitamin D status from 2008 to 2014 was conducted using national data [12]. According to the results of this study, the confirmed prevalence of vitamin D deficiency using the same criteria as in the study referenced above was 65.7% in males and 76.7% in females. Additionally, the change in prevalence between 2008 and 2014 showed a marked increase in vitamin D deficiency from 51.8% to 75.2% in males and from 68.2% to 82.5% in females. Our aim was to determine whether the COVID-19 lockdown affected serum 25(OH)D levels in Koreans. We also aimed to identify changes in the prevalence of vitamin D deficiency during the COVID-19 era.

## 2. Materials and Methods

We included participants 19 years old or older who underwent medical examinations from March 2018 to November 2021 at a university hospital medical examination center in Korea. To determine the impact of the COVID-19 pandemic on 25(OH)D levels, data from subjects who had serum 25(OH)D tests both before and after the outbreak were analyzed. The period before the outbreak of COVID-19 was set from 1 March 2018 to 31 November 2019. To confirm the assumed effect of reduced sunlight exposure due to outdoor activity restrictions, the beginning of the period after the outbreak of COVID-19 was set at 1 June 2020, about three months after the COVID-19 pandemic was first declared. The end point was set at 31 November 2021 in this study. Individual vitamin D levels vary by season [14], so only those who were tested during the same season before and after the pandemic were included. Seasons were classified as spring (March–May), summer (June–August), autumn (September–November), and winter (December, January–February). A total of 1483 subjects were selected for this study.

The survey subjects filled out a self-report questionnaire from which we obtained data on their medical history, smoking status, and physical activity habits. The age distribution was divided into 4 groups: 19–34 years old, 35–49 years old, 50–64 years old, and 65 years and older. Smoking status was divided into non-smokers, ex-smokers, and current smokers. Subjects were assigned to one of five physical activity categories: sedentary, light activity, moderate activity, vigorous activity, and very vigorous activity. Sedentary behavior is a state in which a person spends an extensive amount of time lying down or sitting, and light activity is a state of being engaged in office work or walking for less than two hours a day. Those deemed to have moderate activity include manufacturing and service industry workers and those who walk two to four hours a day. Participants with vigorous activity include those with high-activity occupations such as those in agriculture, fishery, civil engineering, and construction. Lastly, the very vigorous activity category included athletes and those who perform strenuous activities such as wood-carrying. Using the subjects’ medical histories, we identified those with chronic liver disease, chronic renal disease, or osteoporosis or osteopenia. Those taking hormone replacement therapy or contraceptive pills were also accounted for, as these treatments may affect circulating vitamin D levels.

Peripheral venous blood samples obtained after at least 8 h of fasting were collected in tubes containing liquid ethylenediaminetetraacetic acid (EDTA) and were centrifuged. Serum 25(OH)D was measured using a Chemiluminescent Microparticle Immunoassay (CMIA) using the ARCHITECT 25-OH Vitamin D reagent and ARCHITECT I2000 SR instrument manufactured by Abbott. Vitamin D status was defined in three groups by serum 25(OH)D concentration: deficiency (<20 ng/mL), insufficiency (20–29.9 ng/mL), and sufficiency (≥30 ng/mL) [28].

Chi-square tests were performed to compare the subjects’ medical history, smoking history, and physical activity level before and during the COVID-19 pandemic. Paired *t*-tests were applied to compare body weight, BMI, BP, and 25(OH)D level. Referring to past studies which showed that 25(OH)D levels were gender-dependent [12,29], analyses were performed separately for males and females. The analyses proceeded with the exclusion of missing values. Statistical analyses were performed using the SPSS program (ver. 21.0; IBM Corp., Armonk, NY, USA). The statistical significance level was set at *p* < 0.05.

## 3. Results

### 3.1. Participant Characteristics

Of the 1483 participants, 820 (55.3%) were male and 663 (44.7%) were female (Table 1). Most of the subjects were between the ages of 35 and 64 years. Most of the current smokers were males; 263 (32.3%) of all males were smokers before the outbreak of COVID-19. This number was reduced slightly to 249 (30.4%) after the outbreak of COVID-19. The light physical activity group was the most common of the five, with such activity present in 61.3–66.6% of both males and females (Appendix A).

Most participants’ health checkups were performed from spring to autumn; the fewest number of checkups occurred in winter (Appendix A). In total, 96.8% of males and 98.6% of females were tested in spring, summer, or autumn. Medical history conditions that may have affected vitamin D metabolism and status numbered 34 (4.2%) in males and 35 (5.4%) in females before the pandemic (*p* < 0.001). The prevalence of these conditions in the COVID-19 era increased to 40 (4.9%) in males and 47 (7.1%) in females. This change is statistically significant (*p* < 0.001). Chronic liver disease was the most common relevant condition in males; its prevalence increased by 18.2% from 22 (2.7%) before COVID-19 to 26 (3.2%) after COVID-19. In females, osteoporosis or osteopenia was the most common condition; the prevalence increased by 55.6% from 18 (2.7%) before the onset of COVID-19 to 28 (4.2%) after the onset.

### 3.2. Comparison of 25(OH)D Levels before and after the COVID-19 Pandemic

The overall mean 25(OH)D level was 21.4 ± 10.2 ng/mL in the period before COVID-19 (Table 2). However, it increased to 23.6 ± 11.8 ng/mL during the COVID-19 lockdown (*p* < 0.001). In males, the mean 25(OH)D level increased from 21.8 ± 8.9 ng/mL to 23.4 ± 10.6 ng/mL in the COVID-19 era (*p* < 0.001). In females, the mean increased from 21.0 ± 11.6 ng/mL to 23.9 ± 13.1 ng/mL (*p* < 0.001); this increase was larger than that observed in males (Figure 1).

25(OH)D levels after COVID-19 significantly increased with age, especially in females. In females, the level in the 35–49-year-old group increased from 19.6 ± 10.2 ng/mL to 21.8 ± 11.8 ng/mL (*p* < 0.001). The 50–64-year-old group showed an even larger increase from 23.9 ± 13.6 ng/mL to 27.3 ± 13.8 ng/mL (*p* < 0.001). The highest increase was reported in those 65 or older, from 28.8 ± 12.3 ng/mL to 37.7 ± 18.6 ng/mL (*p* = 0.008).

Table 3 shows the seasonal serum 25(OH)D levels before and after the COVID-19 lockdown. Before the outbreak of COVID-19, the serum 25(OH)D level was the highest at 22.35 ± 9.31 ng/mL in summer (June–August) and the lowest at 20.32 ± 11.26 ng/mL in spring (March–May). However, after the COVID-19 lockdown, the highest mean serum 25(OH)D level (25.76 ± 15.55 ng/mL) was confirmed in spring. Even after the COVID-19 lockdown, the mean serum 25(OH) level was still the lowest in winter (21.27 ± 9.96 ng/mL). The season with the largest change before and after the COVID-19 lockdown was spring (from 20.32 ± 11.26 ng/mL to 25.76 ± 15.55 ng/mL, *p* < 0.001), followed by summer (from 22.35 ± 9.31 ng/mL to 24.22 ± 10.56 ng/mL, *p* <0.001).

### 3.3. Changes in Vitamin D Status after the COVID-19 Pandemic

The prevalence of vitamin D deficiency was highest before and after the COVID-19 pandemic, regardless of gender. However, it was higher in females than in males (Table 4, Figure 2). The pre-COVID-19 lockdown status of vitamin D deficiency among males was 397 out of 820 (48.4%), and among females, it was 378 out of 663 (57.0%). During the lockdown period, vitamin D deficiency status was determined in 361 (44.5%) males and 301 (46.0%) females and decreased by 3.9%p (*p* = 0.005) and 9.0%p (*p* < 0.001), respectively.

Both before and after the COVID-19 pandemic, those with sufficient vitamin D levels comprised the smallest proportion of the population. In males, 133 (16.2%) were vitamin D sufficient before the onset of COVID-19. After the onset of COVID-19, this prevalence increased by 6.4%p to 22.6% (*p* = 0.005). Prior to COVID-19, the prevalence of vitamin D sufficiency in females was 18.3%. However, it increased by 7.4%p (*p* < 0.001) to 25.7% after COVID-19. The degree of this increase was higher than that of males.

## 4. Discussion

The lockdown policies implemented worldwide due to the COVID-19 pandemic not only restricted movement between countries, but also reduced person-to-person exchanges and outings within each country. The streets were less populated, and exposure to sunlight from outdoor activities was assumed to have decreased compared to before the outbreak of COVID-19. However, according to the results of our study, serum 25(OH)D levels increased after the containment policy was implemented. Before the outbreak of COVID-19, the annual total sunshine hours were 2498.2 h, as reported in the 2019 Meteorological Annual Report of the Korea Meteorological Administration. In comparison, during the COVID-19 pandemic, the total annual sunshine hours decreased slightly to 2385.6 h in 2020 and 2478.3 h in 2021 [30]. The total amount of sunlight exposure per year does not explain our findings.

Because Korea is located between 33.12 and 38.58 degrees north latitude, the four seasons tend to show distinct differences. In Korea, the duration of sunshine varies depending on the season, and although it varies from year to year, the longest sunshine durations usually occur in spring. In 2019, before the outbreak of COVID-19, and in 2020–2021, after the outbreak, the sunshine durations were the longest in spring. The seasons with the shortest sunshine durations were winter 2019, summer 2020, and autumn 2021, which varied from year to year [30]. The degree of serum 25(OH)D synthesis through the skin may differ depending on the amount and duration of sunlight exposure. Therefore, a subject’s vitamin D levels may vary from season to season. According to a study on seasonal serum 25(OH)D levels in Sweden, located in the range of 55 to 69 degrees north latitude, serum 25(OH)D levels were highest between July and September [14]. This result was strongly associated with parameters related to sun exposure. However, in our study, the seasonal pattern of serum 25(OH)D levels was found regardless of the amount of sunlight. Although spring is the season with the longest sunshine duration before and after the outbreak of COVID-19, serum 25(OH)D levels before COVID-19 were lowest in spring and highest in summer. After COVID-19, the serum 25(OH)D levels were lowest in winter despite the shortest sunshine duration being in summer and autumn. In relation to these results, it can be inferred that factors other than sunshine duration affected the increase in serum 25(OH)D levels.

Regarding the correlation between the effects of vitamin D in the body and the seasons, studies related to the seasonality of vitamin D receptor expression have also been reported. In a study analyzing peripheral blood mononuclear cells of children enrolled in the BABYDIET cohort in Germany, the expression of the seasonal gene anti-inflammatory circadian transcription factor (ARNTL) was the lowest in winter and the highest in summer [31]. In addition, it was confirmed that the expression of the vitamin D receptor (VDR) was also maximized in summer (June–August). These findings suggest that the immune function of vitamin D is expressed more effectively in summer. However, further research is needed to investigate whether seasonal vitamin D differences can actually improve immune function and lower COVID-19 infection rate.

As COVID-19 spread rapidly around the world, people became more interested in health promotion. The results of numerous studies on the benefits of the intake of vitamin D and other nutritional supplements, particularly as related to respiratory infection prevention, have been cited and reported [25,26,32,33,34,35]. Although the results of these studies are inconsistent, many people have taken vitamin D supplements to prevent COVID-19 infection and stop it from worsening into more serious conditions. In some countries, the government and public health agencies issued recommendations for vitamin D intake, resulting in increased sales of vitamin D products [36,37]. In Korea, vitamin D product sales increased during the COVID-19 pandemic [38,39]. Systematic reviews and meta-analyses performed after the COVID-19 pandemic show that, in addition to vitamin D, there has also been a growing interest in other vitamins and multi-micronutrients related to the prevention of respiratory tract infections [40,41,42]. These papers analyzed several studies of vitamins A, B, C, D, E, and other minerals. They found that vitamin and mineral supplementation to prevent respiratory tract infections appears to be ineffective or limited due to a lack of definitive data and inconsistent results. However, there is still a growing interest in the effects of taking these supplements, so further research is required.

Osteoporosis and osteopenia are more common in females, and their prevalence has increased significantly in females since the COVID-19 pandemic began. One reason for this is that the diagnosis rate of osteoporosis and osteopenia increases with age, and their prevalence worldwide is gradually increasing each year [27]. In addition, consistent with our results, osteoporosis has a significantly higher prevalence in females than in males. Prescribing vitamin D supplements is often included in osteoporosis treatment programs. This is especially true because vitamin D deficiency tends to become worse with each passing year [12]. For these reasons, oral intake of vitamin D may have increased more in females than males, especially in older females. This may have been the reason for the increase in the mean 25(OH) level since the outbreak of COVID-19, especially in older females.

In addition, during the COVID-19 lockdown period, individuals’ sports participation also changed. According to the 2020 National Sports Survey conducted in Korea, the participation rate in indoor sports decreased from 66.6% in 2019 before the COVID-19 pandemic to 60.1% in 2020 after the institution of the COVID-19 lockdown policy [43]. However, despite this decline in the use of indoor sports facilities, the participation rate in outdoor sports, such as walking, mountaineering, cycling, and golf, increased. A minority of individuals may have experienced increased exposure to sunlight because of this.

One of the limitations of this study is that it does not represent the entire population because it targeted adults who received a comprehensive health checkup at a single institution. Furthermore, exclusions included those who postponed or canceled their health check-up appointments for fear of COVID-19 infection. If not for fear of COVID-19 exposure, many more people with significant health concerns would have participated. Another limitation was that, due to the retrospective design, there was insufficient investigation into the relationship between osteoporosis, osteopenia, and vitamin D intake status. Therefore, a causal relationship between vitamin D intake and increase in serum 25(OH)D levels could not be established. Nevertheless, the most important strength of this study is that it is the first to investigate changes in vitamin D status before and after the COVID-19 pandemic in the same sample of Korean adults. We also noted that there are seasonal differences in 25(OH)D levels in individuals, which may be confounding factors in comparing its levels before and after the COVID-19 pandemic. To reduce the effect of these confounding factors, only 25(OH)D levels measured in the same season before and after the pandemic in one individual were compared.

## 5. Conclusions

The mean 25(OH)D level increased in both male and female Korean adults in the COVID-19 era. Correspondingly, the prevalence of vitamin D deficiency decreased. This suggests that even though social activities and outings decreased due to the COVID-19 lockdown, people’s health interest and efforts increased. Further research is needed on changes in 25(OH)D levels due to the COVID-19 lockdown, excluding the effects of vitamin D supplements. In addition, research into whether elevated serum 25(OH)D levels reduce the rate of COVID-19 infection and disease severity could provide valuable information.

## Figures and Tables

**Figure 1 nutrients-14-04863-f001:**
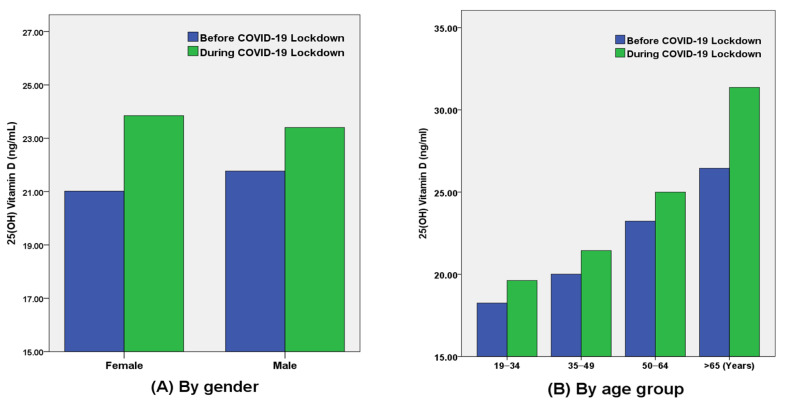
Comparison of 25(OH)D levels before and after the COVID-19 pandemic. These were compared by (**A**) gender and (**B**) age group.

**Figure 2 nutrients-14-04863-f002:**
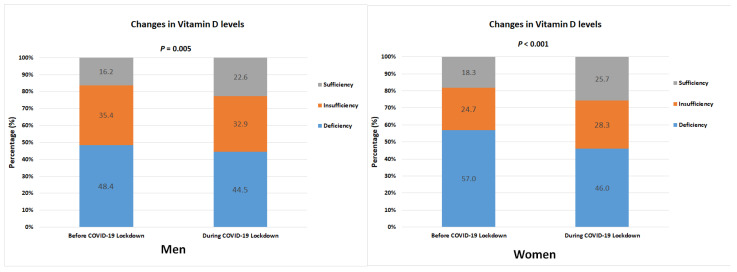
Changes in the prevalence of deficiency (<20 ng/mL), insufficiency (20–29.9 ng/mL), and sufficiency (≥30 ng/mL) of vitamin D by gender after the COVID-19 pandemic.

**Table 1 nutrients-14-04863-t001:** Demographic Characteristics of Participants.

	Before COVID-19 Lockdown (*n* = 1483)			During COVID-19 Lockdown (*n* = 1483)		
	Male (*n* = 820)	Female (*n* = 663)	*p*-Value	Male (*n* = 820)	Female (*n* = 663)	*p*-Value
Age (year)			<0.001			<0.001
19–34	50 (6.1)	92 (13.9)		38 (4.6)	76 (11.5)	
35–49	346 (42.2)	337 (50.8)		294 (35.9)	294 (44.3)	
50–64	372 (45.4)	211 (31.8)		423 (51.6)	259 (39.1)	
≥65	52 (6.3)	23 (3.5)		65 (7.9)	34 (5.1)	
Weight (kg)	74.8 ± 11.4	58.7 ± 9.0	<0.001	74.4 ± 11.3	58.9 ± 9.3	<0.001
BMI (kg/m^2^)	25.3 ± 3.3	23.1 ± 3.5	0.065	25.2 ± 3.3	23.2 ± 3.6	0.008
Smoking			<0.001			<0.001
Non-smoker	216 (26.5)	640 (97.1)		217 (26.5)	639 (96.7)	
Ex-smoker	335 (41.2)	11 (1.7)		352 (43.0)	12 (1.8)	
Current smoker	263 (32.3)	8 (1.2)		249 (30.4)	10 (1.5)	
Physical Activity			0.001			0.005
Sedentary	163 (20.6)	112 (17.6)		155 (19.7)	106 (16.7)	
Light	485 (61.3)	408 (64.1)		483 (61.4)	423 (66.6)	
Moderate	83 (10.5)	96 (15.1)		90 (11.4)	84 (13.2)	
Vigorous	49 (6.2)	17 (2.7)		53 (6.7)	18 (2.8)	
Very vigorous	11 (1.4)	4 (0.6)		6 (0.8)	4 (0.6)	
Blood pressure (mmHg)						
Systolic	126.0 ± 13.8	119.3 ± 14.6	0.089	126.2 ± 13.5	120.4 ± 14.8	0.006
Diastolic	77.2 ± 10.1	72.0 ± 10.1	0.905	79.9 ± 10.0	72.9 ± 10.4	0.392
Season			0.003			0.002
Spring	139 (17.0)	83 (12.5)		140 (17.1)	83 (12.5)	
Summer	265 (32.3)	206 (31.1)		264 (32.2)	206 (31.1)	
Autumn	390 (47.5)	365 (55.0)		390 (47.5)	365 (55.0)	
Winter	26 (3.2)	9 (1.4)		26 (3.2)	9 (1.4)	
Past History			<0.001			<0.001
Osteoporosis or Osteopenia	8 (1.0)	18 (2.7)		9 (1.1)	28 (4.2)	
HRT or contraceptive pills	0 (0.0)	11 (1.7)		0 (0.0)	12 (1.8)	
Chronic liver disease	22 (2.7)	5 (0.8)		26 (3.2)	5 (0.8)	
Chronic renal disease	4 (0.5)	1 (0.2)		5 (0.6)	2 (0.3)	

Values are presented as mean ± standard deviation for continuous variables or number (%) for categorial variables. *p*-values are from *t*-tests for continuous variables and chi-squared tests for categorial variables, respectively. BMI, body mass index; HRT, hormone replacement therapy; COVID-19, coronavirus disease.

**Table 2 nutrients-14-04863-t002:** 25(OH)D levels according to gender in different age groups before and during the COVID-19 lockdown period.

	Total (*n* = 1482)			Male (*n* = 820)			Female (*n* = 662)		
	Before COVID-19 Lockdown	During COVID-19 Lockdown	*p*-Value	Before COVID-19 Lockdown	During COVID-19 Lockdown	*p*-Value	Before COVID-19 Lockdown	During COVID-19 Lockdown	*p*-Value
25(OH)D									
Total (ng/mL)	21.4 ± 10.2	23.6 ± 11.8	<0.001	21.8 ± 8.9	23.4 ± 10.6	<0.001	21.0 ± 11.6	23.9 ± 13.1	<0.001
19–34 years	18.3 ± 8.6	20.1 ± 9.6	0.027	19.2 ± 7.2	20.2 ± 7.8	0.291	17.8 ± 9.3	20.0 ± 10.6	0.051
35–49 years	20.0 ± 9.1	21.9 ± 10.5	<0.001	20.4 ± 8.0	22.1 ± 9.2	<0.001	19.6 ± 10.2	21.8 ± 11.8	<0.001
50–64 years	23.2 ± 11.1	25.5 ± 12.5	<0.001	22.9 ± 9.5	24.4 ± 11.6	0.006	23.9 ± 13.6	27.3 ± 13.8	<0.001
≥65 years	26.5 ± 10.9	31.1 ± 15.1	0.005	25.4 ± 10.3	28.1 ± 12.3	0.141	28.8 ± 12.3	37.7 ± 18.6	0.008

Values are presented as mean ± standard deviation. *p*-values are for differences in serum 25(OH)D levels before and after the outbreak of COVID-19 by gender and age distribution. 25(OH)D, 25-hydroxyvitamin D; COVID-19, coronavirus disease.

**Table 3 nutrients-14-04863-t003:** Seasonal serum 25(OH)D levels before and during the COVID-19 lockdown period.

Season	25(OH)D (ng/mL) Before COVID-19 Lockdown	25(OH)D (ng/mL) During COVID-19 Lockdown	*p*-Value
Spring (March–May)	20.32 ± 11.26	25.76 ± 15.55	<0.001
Summer (June–August)	22.35 ± 9.31	24.22 ± 10.56	<0.001
Autumn (September–November)	21.26 ± 10.35	22.69 ± 11.22	<0.001
Winter (December, January–February)	20.42 ± 11.84	21.27 ± 9.96	0.709

Values are presented as mean ± standard deviation. *p*-values are for differences in serum 25(OH)D levels before and after the outbreak of COVID-19 by season. 25(OH)D, 25-hydroxyvitamin D; COVID-19, coronavirus disease.

**Table 4 nutrients-14-04863-t004:** The rates of 25(OH)D deficiency, insufficiency, and sufficiency before and during COVID-19 lockdown periods.

			Before COVID-19 Lockdown	During COVID-19 Lockdown	
			*n* (%)	*n* (%)	*p*-Value
Male			820	811	0.005
	25(OH)D	Deficiency	397 (48.4)	361 (44.5)	
		Insufficiency	290 (35.4)	267 (32.9)	
		Sufficiency	133 (16.2)	183 (22.6)	
Female			663	654	<0.001
	25(OH)D	Deficiency	378 (57.0)	301 (46.0)	
		Insufficiency	164 (24.7)	185 (28.3)	
		Sufficiency	121 (18.3)	168 (25.7)	

*p*-values are for differences in serum vitamin D status before and after the outbreak of COVID-19 in males and females. 25(OH)D, 25-hydroxyvitamin D; COVID-19, coronavirus disease.

## Data Availability

Data are available upon reasonable request.

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
