# Peer review of "Changes in Vitamin D Status in Korean Adults during the COVID-19 Pandemic"

_nutrients, 2022, doi:10.3390/nu14224863_

Round 1
Reviewer 1 Report
Interesting topic but there are major issues with this manuscript:
- Subjects evaluated before and after COVID pandemic are not the same so direct comparisons are impossible.
- Demographic characteristics of participants have not been compared between subjects evaluated before and after COVID pandemic. This is of primary importance as confounding factors (i.e. imbalance between groups) may influence the results
- No information about vitamin D intake is available
- Major confounding variables have not been taken into account (sun exposure, comorbidities, supplementation intake
- The use of paired t-test is a problematic since that statistic is only used when each subject is measured twice. Clearly, this is not the case here.
Author Response
Response to Reviewer 1 Comments
Point 1: Subjects evaluated before and after COVID pandemic are not the same so direct comparisons are impossible.
Response 1: Thank you for your interest and advice on this paper. This study was conducted on 1483 adults who underwent a serum 25(OH)D test at a screening center at a university hospital in South Korea. This study compared serum 25(OH)D levels before and after the onset of COVID-19 in these 1483 same subjects. In other words, it is a study comparing the serum 25(OH)D level before and after the onset of COVID-19 in each participant in the same group, and we consider this to be a significant advantage of this study. This study is a comparative study of the same subjects before and after the COVID-19 pandemic.
Point 2: Demographic characteristics of participants have not been compared between subjects evaluated before and after COVID pandemic. This is of primary importance as confounding factors (i.e. imbalance between groups) may influence the results.
Response 2: As I answered above, this study identified changes before and after COVID-19 in the same subjects. Thus, demographic characteristics of participants represent changes before and after the COVID-19 outbreak within the same population of participants.
Point 3: No information about vitamin D intake is available.
Response 3: That's a very exact point. This study is a retrospective study and does not include an intentional investigation of vitamin D intake. In Korea, many people do not think of vitamin D intake as a drug intake, but as a supplement, so they do not talk about whether or not to take it during the medication survey. However, we confirmed osteoporosis or osteopenia during past history and can only infer that some of these people will take vitamin D supplements. In this section, we have further described the limitations of the discussion section. Thank you for your attentive comments.
Point 4: Major confounding variables have not been taken into account (sun exposure, comorbidities, supplementation intake.
Response 4: Controlling for the main confounding variables was an important issue for us. To control for seasonal differences in sunlight that could affect serum 25(OH), we selected participants only from those who had a serum 25(OH)D test in the same season before and after COVID-19 outbreak. For example, people who were tested in the spring (March-May) before the COVID-19 pandemic also tested in the spring (March-May) after the COVID-19 pandemic. And we compared these two results. The other seasons were compared in the same way.
Investigation of past history that may affect vitamin D levels before and after COVID-19 was also needed. We investigated osteoporosis/osteopenia, HRT/taking contraceptive pills, chronic liver disease, and chronic renal disease. The results can be seen in Table 1. Although the prevalence of these diseases is somewhat different, most of them have shown an increasing pattern after the COVID-19 pandemic.
Also, as answered in response 3 above, there was a limit to confirming whether supplements were taken. It appears that vitamin D supplementation apparently had an effect on serum 25(OH)D levels. After the COVID-19 outbreak, the results of this study show an increase in serum 25(OH)D levels, despite reduced exposure to sunlight due to restrictions on outdoor activities and quarantine measures. We speculate that this is due to increased intake of vitamin D supplements. Additionally, increased sales of vitamin D supplements support this reasoning.
We have mentioned these in the methods and discussion sections of the study.
Point 5: The use of paired t-test is a problematic since that statistic is only used when each subject is measured twice. Clearly, this is not the case here.
Response 5: Since we measured and compared serum 25(OH)D before and after the COVID-19 pandemic in the same subjects, we thought the paired t-test was the best option.
Reviewer 2 Report
I think that the study provides interesting insights.
I have the following major comments that should be addressed to improve the quality of the study
- I think that the authors should employ adjusted models to test their findings. In particular, mixed effects models would be appropriate to test which predictors are associated with the increase of vitamin D levels. This would markedly increase the quality of the study and the study design fits well with this type of analysis
- Were there missing data? How were they managed?
- Are there changes in vitamin D supplementation? I wonder if some subjects with low levels of vitamin D underwent vitamin D supplementation to increase their vitamin D status
- Are data on diet available?
- Differences in the seasonality of the tests should be considered within the analyses
- I think that the discussion would improve if the authors briefly addressed new insights on vitamin D. In particular, recent evidence on the different effects of vitamin D in the different seasons should be addressed (please consider PMID: 25965853)
- I would also briefly address the role of other vitamins and micronutrients that were claimed and studied during the pandemic (please consider PMID: 33916257, PMID: 33435749, PMID: 34626488) within the discussion
Author Response
Response to Reviewer 2 Comments
Point 1: I think that the authors should employ adjusted models to test their findings. In particular, mixed effects models would be appropriate to test which predictors are associated with the increase of vitamin D levels. This would markedly increase the quality of the study and the study design fits well with this type of analysis.
Response 1: I appreciate your attentive comment. I totally agree with your opinion. In order to use mixed effects models, as you mentioned, we had to know if our subjects were taking Vit D supplements and, if so, at what doses they were taking them. Unfortunately, however, these details were not known because our study was a retrospective study using existing medical records. In general, during the COVID-19 lockdown period, it was expected that the vitamin D levels of the study subjects would decrease because they were unable to freely engage in outdoor activities or going out, but it was found that the vitamin D levels increased unexpectedly. We speculated that these results might be related to the sharp increase in vitamin D supplement sales during the COVID-19 lockdown period. Through additional prospective studies in the future, it is necessary to accurately determine whether vitamin D supplements are taken and the dosage, and whether elevated vitamin D levels affect the reduction of the rate and severity of COVID-19 infection. We have further described these in the conclusion section, as well as the study limitations in the discussion section.
Point 2: Were there missing data? How were they managed?
Response 2: Thank you for your thoughtful comments. Since there were very few cases of missing data among the data of 1483 subjects of the total study, we excluded missing data from each data analysis process. We have further commented on this point in statistical analysis.
Point 3: Are there changes in vitamin D supplementation? I wonder if some subjects with low levels of vitamin D underwent vitamin D supplementation to increase their vitamin D status.
Response 3: This study is a retrospective study and does not include an intentional investigation of vitamin D intake. In Korea, many people do not think of vitamin D intake as a drug intake, but as a supplement, so they do not talk about whether or not to take it during the medication survey. However, we confirmed osteoporosis or osteopenia during past history and can only infer that some of these people will take vitamin D supplements.
Point 4: Are data on diet available?
Response 4: Unfortunately, we don't have data on the participants' diets or types of meals. This is because this study is a retrospective study. Therefore, it is not possible to determine whether there are changes in the diet that would increase vitamin D levels.
Point 5: Differences in the seasonality of the tests should be considered within the analyses.
Response 5: Your point is correct. Because South Korea is located between 33.12 and 38.58 degrees north latitude, the characteristics of the four seasons repeat every three months. Therefore, vitamin D levels of study subjects may appear differently depending on the time of the test. In order to control such variables, all subjects were selected from those who were tested in the same season before and after the COVID-19 outbreak. For example, people who were tested in the spring (March-May) before the outbreak of COVID-19 were tested in the spring (March-May) after the outbreak of COVID-19. The other seasons were compared in the same way. This was to control for seasonal differences in serum 25(OH)D levels that may occur between individuals. This study was to investigate the change by comparing the average serum 25(OH)D level before and after the COVID-19 pandemic in the same subject group after adjusting for seasonal changes in individual serum 25(OH)D levels. Therefore, in this study, the mean serum 25(OH)D was used as it was without considering seasonal differences. We have additionally presented Vit D levels for each season in the discussions. Again, thank you for your attentive comment.
Point 6: I think that the discussion would improve if the authors briefly addressed new insights on vitamin D. In particular, recent evidence on the different effects of vitamin D in the different seasons should be addressed (please consider PMID: 25965853).
Response 6: The references you provided were new and informative for us. Seasonal differences in vitamin D receptor expression and information on the seasonal gene ARNTL could provide new insights into vitamin D-related research. We further analyzed seasonal differences in serum 25(OH)D levels and their changes since the COVID-19 pandemic and added the following content and references to the discussion.
à As for the correlation between the effects of vitamin D in the body and the seasons, studies related to the seasonality of vitamin D receptor ex-pression have also been reported. In a study analyzing peripheral blood mononuclear cells of children enrolled in the BABYDIET cohort in Germany, the expression of the seasonal gene anti-inflammatory circadian transcription facotr (ARNTL) showed the lowest value in winter and the highest value in summer [50] ######. In addition, it was confirmed that the expression of the vitamin D receptor (VDR) was also maximized in summer (June-August). According to this, it can be thought that the immune function of vitamin D is expressed more effectively in summer. However, further research is needed to investigate whether seasonal vitamin D differences actually improved immune func-tion and lowered COVID-19 infection rate.
Point 7: I would also briefly address the role of other vitamins and micronutrients that were claimed and studied during the pandemic (please consider PMID: 33916257, PMID: 33435749, PMID: 34626488) within the discussion.
Response 7: Your comments and additional references have been very helpful. We added references and discussions on the preventive effect of not only vitamin D but also other supplements on respiratory tract infection, and inserted the following content into the discussion.
à A systematic review and meta-analysis of several types of vitamins and multiple micronutrients was conducted in relation to the prevention of respiratory tract infections. These papers analyzed several studies of vitamins A, B, C, D, E and several other minerals. In conclusion, vitamin and mineral supplementation to prevent respiratory tract infections appears to be ineffective or limited due to the lack of definitive data and the inconsistent results.
Round 2
Reviewer 1 Report
I am fine with the comments of the authors